# Position: There Are Futures That Benchmark-Driven AI Cannot See

Sobhan Lotfi [* 1]  Ava Iranmanesh [* 2]  Lachin Naghashyar [3]  Ali Shirali [4]  Fateme Nateghi Haredasht [5]
Sanmi Koyejo [5]  Philip Torr [6]  Yong Suk Lee [7]  Fazl Barez [6]  Joel Lehman [8]  Peter Norvig [5 9]  Arvind Narayanan [10]

## Abstract

In biology, traits evolved for one function sometimes become decisive for another; this is exaptation, and scientific progress works the same way. AI's benchmark-centered selection environment taxes exaptation: when one selection rule dominates, ideas that do not fit it have nowhere to persist. The cost grows acute as the field shifts from asking *can machines exhibit intelligent behavior?* to asking *can they do so while remaining aligned, interpretable, and safe?* These are philosophically distinct questions that may require discoveries we cannot specify. We propose mechanisms to restore exaptive capacity without abandoning benchmarking: plural evaluation regimes, protected venues for non-comparable work, long-horizon funding, and training norms that encourage researchers to question selection rules, not only optimize within them.

## 1. Introduction

> *"Live long and be patient; this wheel of tricks has a thousand wilder games to play."*
>
> — Hafez Shirazi

Modern machine learning is largely organized around benchmark performance. This has proven to be an unusually effective epistemic choice: shared tasks, datasets, and metrics have made progress directly comparable across labs, legible to funders, and easier to coordinate as the field expanded (Jelinek, 1997; Harman, 1993). Rather than resolving longstanding philosophical debates about the nature of intelligence, benchmarking offered a pragmatic alternative: measuring what systems can objectively do (Koch & Peterson, 2024). We do not argue that benchmarking was a mistake; we ask what is lost when evaluation is concentrated around a single selection rule. Any dominant criterion privileges what it can measure while sidelining what it cannot; the resulting omissions are easy to overlook because the system otherwise appears to be working. As Kuhn noted, periods of "normal science" routinely set aside anomalies; those anomalies can later become central when a paradigm enters crisis and shifts (Kuhn, 1962; Arthur, 1996).

Our lens is exaptation, a biological term for traits evolved for one function that later become decisive for another (Gould & Vrba, 1982). In research ecosystems, ideas can similarly acquire value in ways their originators did not anticipate (Wimsatt, 2007; Stanley & Lehman, 2015). Crucially, exaptation depends on persistence: ideas with latent potential must survive periods when they appear uncompetitive by prevailing metrics in order to later realize that potential. **We argue that benchmark-centered evaluation imposes an exaptation tax by reducing the survival of work whose value is latent, indirect, or only revealed under a future problem regime.** In this sense, benchmarks do not merely evaluate research; they *organize* it, shaping who can participate, what kinds of contributions are rewarded, and which questions and methods come to seem fundable, publishable, and legitimate (Bourdieu, 1975).

The stakes of this tradeoff rise as the field's guiding question changes. For much of its history, the implicit target was **Q1**: can machines exhibit intelligent behavior? Increasingly, the focus has shifted toward **Q2**: can machines exhibit intelligent behavior such that they are aligned, interpretable, safe, and governable? These "such-that" clauses are not simply additional constraints on the original question. They change what counts as success: Q1 has ground truth that exists in the world independently of who evaluates the system; Q2 does not. Whether a system is aligned, interpretable, or safe depends on criteria that must be constructed through deliberation rather than read off from the task itself (Russell & Norvig, 2021; Gabriel, 2020). The field has not yet established that the correlation between benchmark performance

---

[*]Equal contribution  [1]Department of Computer Engineering, Sharif University of Technology, Tehran, Iran [2]Pennsylvania State University, State College, PA, USA [3]Microsoft, London, UK [4]University of California, Berkeley, CA, USA [5]Stanford University, Stanford, CA, USA [6]University of Oxford, Oxford, UK [7]Keough School of Global Affairs, University of Notre Dame, Notre Dame, IN, USA [8]Lila Sciences, San Francisco, CA, USA [9]Google, Mountain View, CA, USA [10]Princeton University, Princeton, NJ, USA. Correspondence to: Ava Iranmanesh <ami5520@psu.edu>.

*Proceedings of the $43^{rd}$ International Conference on Machine Learning*, Seoul, South Korea. PMLR 306, 2026. Copyright 2026 by the author(s).

and real progress that held for Q1 holds for Q2. Importing Q1's evaluation framework into Q2 without validating that it transfers is not a conservative choice; it forecloses the exploratory work from which any adequate answer to Q2 would have to emerge.

The remainder of the paper traces how benchmark-centered evaluation became dominant, showing that its current centrality is historically contingent rather than inevitable (Section 2); diagnoses the mechanisms by which it governs research trajectories to clarify where intervention is possible (Section 3); develops the such-that shift as a problem that differs from Q1 not in degree but in kind (Section 4); and proposes institutional mechanisms for restoring exaptive capacity without abandoning benchmarking (Section 5). The cost of getting this wrong is not a missed benchmark; it is a field that optimizes confidently toward futures it can measure, and misses the ones it needs most.

**Conflict of Interest Disclosure:** Lachin Naghashyar is employed by Microsoft. Joel Lehman is employed by Lila Sciences. Peter Norvig is employed by Google. The authors declare that this research was conducted in the absence of any commercial or financial relationships that could be construed as a potential conflict of interest.

# 2. Background: Exaptation and AI History

This section reviews exaptation and the history of AI evaluation in order to show how selection regimes shape which ideas persist long enough to matter.

## 2.1. Exaptation and the Persistence of Ideas

Exaptation was formally introduced by Gould and Vrba as a refinement of adaptation, referring to traits that evolved for one function and were later co-opted for another (Gould & Vrba, 1982). Exaptation depends on persistence beyond a trait's original function. Such traits may be adaptive in their original role, but that fitness does not explain their later significance. In early mammals, for example, bones that once played a role in jaw mechanics were later repurposed into the middle ear, enabling hearing (Anthwal et al., 2013). Their importance in this new role was not explained by optimization for the original one, but by their survival long enough to be reused. This logic has been extended beyond biology to scientific and technological change, where ideas or tools developed for one problem can later become decisive for another, provided they persist through periods in which their value lies outside what prevailing evaluative standards reward (Ferreira et al., 2020). Magnetrons developed for radar became the basis for microwave cooking. Non-Euclidean geometry, developed by Riemann in 1854 as pure mathematics with no physical application, became essential to general relativity sixty years later. Exaptation thus highlights a mode of progress in which long-term innovation depends not only on optimization for current objectives, but on the continued institutional viability of ideas whose value is not yet legible.

## 2.2. Evaluation Regimes in the History of AI

In the pre-benchmark era of AI (1956–1987), the field developed as a pluralistic but loosely evaluated research space, with competing theories of intelligence and no shared standards for evaluating progress (Russell & Norvig, 2021). The Dartmouth proposal articulated an expansive ambition, asserting that "every aspect of learning or any other feature of intelligence can in principle be so precisely described that a machine can be made to simulate it", without specifying how progress should be assessed across approaches (McCarthy et al., 1955). This openness allowed symbolic, connectionist, and heuristic methods to coexist, but left no shared standard for resolving disputes. Early AI has been characterized as a form of "protoscience," in which organic evaluation mechanisms proved insufficient to stabilise consensus, contributing to recurring hype–disappointment cycles and the AI winters of the 1970s and late 1980s.

In the benchmark intervention period (1987–2012), major funders—most notably DARPA—helped reorient AI away from resolving foundational disputes about intelligence and toward measurable progress on well-specified tasks with clear evaluation metrics. This shift crystallised into the *Common Task Framework*, which organized research around shared datasets, held-out evaluation, and automated scoring, enabling direct comparison among methods on a fixed task. In this sense, benchmarking served as a pragmatic bypass, making progress legible through quantitative performance (Koch & Peterson, 2024; Raji et al., 2021a).

From roughly 2012 onward, AI entered a period of extraordinary coherence driven by the convergence of large datasets, GPU acceleration, and accumulated advances in learning algorithms. Following the 2012 ImageNet breakthrough, deep learning's success reflected favourable positional factors—large labelled datasets and GPU acceleration—rather than a settled theory of intelligence (Krizhevsky et al., 2012). Once this combination proved decisive across vision, speech, and language tasks, scaling model size, data, and compute emerged as a reliable and comparatively low-risk path to performance gains. Empirical scaling studies reinforced this strategy by demonstrating predictable improvements with increased scale, further entrenching optimization over a small set of quantitative benchmarks as the dominant research practice (Hestness et al., 2017). As a result, the field increasingly treated benchmark performance as the main indicator of progress, yielding rapid gains and clear success signals, and producing an evaluation regime tightly focused on predictive accuracy. By mid-2026, leading benchmarks

have largely saturated, making this regime's limits visible rather than merely theoretical.

## 2.3. Exaptation in Modern Machine Learning

The historical record suggests that evaluation regimes play a role in determining which lines of work remain visible, disseminated, and sustained by the research community long enough to matter. Modern machine learning provides concrete illustrations of this pattern: several of its core capabilities emerged through the repurposing of ideas, tools, or structures, sometimes developed for adjacent problems or subfields, and later applied in new roles. At the infrastructural level, graphics processing units (GPUs) provide a canonical example. Originally developed to accelerate real-time graphics rendering, GPUs were optimised for highly parallel arithmetic operations rather than for general-purpose programmability (Owens et al., 2008; Nickolls et al., 2008). This hardware proved decisive for deep learning by making the training of previously impractical neural networks feasible at scale. The 2012 ImageNet breakthrough, enabled by GPU-trained convolutional networks, marked a turning point that accelerated the adoption of deep learning across vision, speech, and language (Krizhevsky et al., 2012). In this sense, GPUs did not constitute a theoretical advance in learning algorithms, but an infrastructural exaptation that reshaped the pace and direction of machine learning research.

Exaptation has also operated at the levels of model architecture and training methodology. The U-Net architecture was originally introduced for biomedical image segmentation (Ronneberger et al., 2015). In later work on denoising diffusion probabilistic models, U-Net–based architectures were repurposed as the core backbone for generative modeling, where the same multi-scale, skip-connected structure supports iterative noise prediction and removal, enabling high-fidelity image synthesis (Ho et al., 2020).

A useful illustration comes from the history of neural networks in the late 1990s and early 2000s. Support vector machines appeared stronger on the benchmarks of the era, and the field shifted attention accordingly. This was locally rational but ultimately misleading: advances in optimization, data availability, and GPU acceleration later enabled deep networks to become the foundation of modern ML (Bengio et al., 2013; LeCun et al., 2015). The episode illustrates how evaluation regimes shape which ideas are prematurely set aside.

The 2014 paper introducing generative adversarial networks offers a more recent and precisely documented case (Goodfellow et al., 2014). The authors hedged explicitly on sample quality, writing that their results made no claim of superiority over existing generative methods, framing the contribution as demonstrating a framework's potential. One reviewer concluded that the submission was "may not yet be quite strong enough for NIPS." The paper was accepted. It became one of the most cited works in the history of machine learning. What the evaluation regime could not easily measure was the generative framework itself; the conceptual architecture that would later reshape the field.

# 3. Present Diagnosis: The Tyranny of Benchmarks

Benchmarks do not merely evaluate research, they increasingly *organize* it. This governance operates on three levels. First, benchmark-centered progress selects for particular resource profiles (compute, engineering infrastructure, familiarity with standardized datasets), shaping who can participate and who can sustainably stay. Second, benchmarks shape what even the selected participants work on, systematically rewarding "hill-climbable" contributions while raising the cost of paradigm-challenging work. Third, benchmarks codify present understanding, while many breakthrough contributions derive their value from *future exaptations*, uses and recombinations that simplistic evaluation cannot anticipate, yet simplistic evaluation determines which lines of work survive long enough to be exapted.

The tyranny is not in any individual benchmark. It is in *benchmark culture* as the field's dominant mode of evaluation: when leaderboards, standardized pipelines, and common-task success become the default currency of legitimacy, they quietly reshape the community, the agenda, and the kind of novelty that can be recognized.

## 3.1. Community Composition

NeurIPS was founded by researchers studying neural information processing systems, a coalition spanning neuroscience, cognitive science, physics, and computer science, unified by interest in how biological and artificial systems process information. That founding coalition has largely moved to the periphery of today's flagship ML venues. Bourdieu offers a framework for understanding such shifts: scientific fields define what counts as legitimate capital, and participants without the right capital learn to *self-disqualify* before explicit rejection occurs (Bourdieu, 1975). In ML today, that cost of entry is not only conceptual; it is often infrastructural.

This narrowing is not incidental to benchmark culture—it is produced by it. Klinger et al. document that AI research has become focused on narrower themes in recent years, and that work involving the private sector tends to be less diverse and more influential than academic work, with specialization toward data-hungry and computationally intensive methods (Klinger et al., 2020; OECD, 2023). A large-scale analysis of 41 million research papers confirms that AI tools amplify individual output while collectively narrowing the

themes researchers pursue (Hao et al., 2026). This concentration also reflects a structural feature of modern AI's funding landscape: unlike earlier technology breakthroughs, from the internet to GPS to semiconductors, which were primarily government- and academia-funded (Mazzucato, 2013), today's AI development is driven by venture capital and large technology companies whose selection pressures favor commercially viable, short-horizon research (OECD, 2023). Benchmark culture amplifies this dynamic by making large-scale compute, data, and engineering infrastructure especially valuable: when progress is measured as incremental gains on shared tasks, the returns to scale in these resources become unusually direct, and compatibility with prevailing hardware and software ecosystems shapes which ideas succeed as much as their scientific merit (Hooker, 2020; Strubell et al., 2019).

Why should we care who participates? One reason is combinatorial: transformative work often emerges from unusual recombinations across methods, domains, and problem framings. The empirical science-of-science literature supports this mechanism at scale. Wu, Wang, and Evans, analyzing 65 million papers and patents, find that small teams more often "disrupt" existing trajectories while large teams more often "develop" and consolidate them (Wu et al., 2019). Uzzi et al. similarly show that the highest-impact science tends to combine highly conventional prior work with a smaller intrusion of atypical combinations (Uzzi et al., 2013). The implication for ML is not that large-scale benchmarked engineering is bad (it is often essential), but that a community whose dominant selection pressures favor scale and consolidation will, at the margin, generate fewer atypical combinations, and atypical combinations are the raw material from which future exaptations are built.

Consider what this means for exaptation in ML itself. The attention mechanism emerged from work at the intersection of machine learning and computational linguistics — a community that maintained distinct evaluation standards and institutional identity for decades (Bahdanau et al., 2014). ACL remains one of AI's oldest venues, and translation one of its oldest problems. That community sustained the conceptual foundations, sequence modeling, alignment, encoder-decoder structure, later extended into the transformer (Bahdanau et al., 2014; Vaswani et al., 2017). AlphaFold2 then adapted these architectures to protein structure prediction, recognized by the 2024 Nobel Prize in Chemistry (Jumper et al., 2021; The Nobel Prize, 2024). The lineage is counterintuitive: solving a grand challenge in structural biology required having had linguists in AI. This was not planned, and could not have been. It was possible because a heterogeneous research ecosystem preserved communities with distinct problem framings long enough for unexpected recombinations to occur. Benchmark culture can sustain narrow subcommunities. But cross-domain exaptations depend

on the broader ecosystem that benchmark monoculture selects against.

This is why "community composition" is not a sociological side note: it is a technical precondition for certain kinds of progress. If benchmark culture disproportionately rewards resource-intensive, pipeline-compatible work, then it indirectly shapes the space of recombinations the field will ever get a chance to discover.

### 3.2. Research Agenda

Section 3.1 established that benchmark culture shapes who participates in ML research. But benchmarks also shape what even selected participants choose to work on. Donoho articulates a compelling positive case: "frictionless reproducibility", the combination of shared data, shared code, and competitive challenges, can accelerate progress by enabling researchers to directly compare, iterate, and build with minimal coordination cost (Donoho, 2024). This is real and valuable. Yet Recht, in his commentary on Donoho, offers a revealing reframe: "We can view machine learning practice as a massively parallel genetic algorithm that fixates on goals where hill climbing is possible and ignores those where it isn't" (Recht, 2024).

This dynamic maps naturally onto Kuhn's distinction between normal science and revolutionary science (Kuhn, 1962). Benchmarks, leaderboards, and standardized pipelines are "normal science with the friction removed": they create extraordinary efficiency at paradigm-consistent progress. But the same infrastructure raises the barrier to work that challenges the paradigm. To depart from the dominant framing, one must often do two things at once: introduce a new idea *and* demonstrate competitive performance on metrics and pipelines optimized for the incumbent paradigm. The asymmetry is structural: mature baselines inherit years of tuned hyperparameters, training recipes, compute infrastructure, and evaluation conventions, while genuinely new approaches must recreate much of that stack merely to become comparable.

The transition from statistical to neural machine translation shows this asymmetry playing out over an entire research generation. Neural approaches existed early (Forcada & Ñeco, 1997), but SMT dominated for decades under benchmark regimes and engineering ecosystems that strongly favored it (Stahlberg, 2020). NMT "arrived" as a benchmark winner only after the broader neural ecosystem (compute, data, training methods, and engineering practices) matured to the point where neural systems could become competitive quickly (Sutskever et al., 2014; Bahdanau et al., 2014). The lesson is not that benchmarks prevented NMT (they eventually validated it). The lesson is that benchmark-driven selection pressures can delay or tax paradigm shifts until the surrounding ecosystem makes them "one-shot competitive."

Recent analyses confirm that benchmarks steer agendas, not merely measure outcomes. Systematic study of sixty LLM benchmarks finds that saturation is now the norm: once models approach ceiling performance, the field's response is to create new benchmarks rather than to question the evaluative regime itself (Akhtar et al., 2026; Ott et al., 2022). Leaderboard rankings have been shown to reflect undisclosed private testing practices that benefit a handful of providers, distorting the playing field that nominally drives research priorities (Singh et al., 2025). Critiques from within ML further emphasize construct validity problems when benchmarks are treated as broad measures of progress (Raji et al., 2021b; Bechler-Speicher et al., 2025). Benchmarks function as selection environments that shape which questions are "worth asking" and which styles of research are "viable careers."

### 3.3. The Limits of Evaluation

Every evaluation instantiates a theory of what matters. That theory is necessarily incomplete because the future problem landscape is not knowable in advance. This is not merely ordinary uncertainty (unknown parameters); it is often closer to *Knightian* uncertainty: qualitative shifts in tasks, environments, and use cases that cannot be exhaustively enumerated today (Knight, 1921). Lehman et al. make this point in a particularly ML-relevant form, arguing that ML systems face a "Knightian blindspot" when confronted with qualitatively novel challenges generated by complex, creative environments (Lehman et al., 2025). When evaluation is optimized for what we can currently specify, it can systematically underweight work whose value depends on what we cannot yet quantitatively specify.

Historical patterns in science and technology recur. Work can be "ahead of its time," recognized only when complementary tools or applications arrive (van Raan, 2004; Garfield, 1980). In AI, the most valuable contributions can be those that later become indispensable components of systems aimed at problems we do not yet know how to write down.

Benchmarks also encode hidden assumptions about deployment. Computer vision benchmarks, for example, historically defined tasks through particular label sets, sensors, and controlled conditions—assumptions that later turned out to be misaligned with robotics, distribution shift, and real-world robustness (Everingham et al., 2010; Scharstein & Szeliski, 2002; Hendrycks & Dietterich, 2019). This does not indict benchmarking; it clarifies its limits: benchmarks measure what they measure.

Path dependence theory formalizes why such limits can have long-run consequences. When a research direction experiences increasing returns (tooling, shared pipelines, network effects in adoption), early advantages compound and alternatives become harder to pursue even if they would be better in retrospect (Arthur, 1989; 1996). Benchmark culture creates increasing returns for specific research profiles: compute, engineering infrastructure, and familiarity with standard datasets. These advantages accumulate. The resulting loss is largely invisible because we cannot observe innovations never pursued. Yet the structural dilemma remains: benchmarks codify present understanding; breakthroughs often require violating that understanding. No quantitative evaluation method can foresee exaptations. But our evaluation methods determine which research survives long enough to be exapted.

## 4. The Such-That Problem

The three dimensions diagnosed in Section 3 (community composition, research agenda, and the limits of evaluation) each become more acute as the field's guiding question shifts from Q1 to Q2, and the worsening is not one of degree. Community composition matters more because such-that problems require contributions from disciplines that benchmark culture systematically marginalizes: philosophy, political science, and the social sciences that have grappled with contested normative questions for decades. Research agenda matters more because benchmark-driven selection cannot orient toward goals that are not yet fully specified; one cannot hill-climb toward a summit whose location remains unknown. The limits of evaluation become the central problem rather than a manageable one: for capability research, the gap between benchmark performance and real progress is imperfect but has been validated by decades of use; for such-that problems, that validation does not exist, and premature convergence on current proxies may foreclose discovery of what we actually need to measure. The following subsections develop why the shift from Q1 to Q2 changes the nature of the problem, not only its difficulty.

### 4.1. The Such-That Problem Is in the Air

We cite what follows as evidence that this shift is already institutional, not as an evaluation of the efforts themselves. The arc of this section is observational: 4.1 documents the shift, 4.2 argues it is different in kind from what preceded it, and 4.3 draws the implication for exaptation.

Evidence suggests a shift in research attention. Turing Award recipients have begun publicly advocating for safety research (Bengio et al., 2024). Major venues have recognized this work: NeurIPS lists safety, interpretability, fairness, and privacy among its topical areas (NeurIPS, 2025b), hosts alignment and interpretability workshops (NeurIPS, 2024b; Abrol et al., 2024), and requires ethics review (NeurIPS, 2025a). Award committees have followed, recognizing alignment and safety work at NeurIPS 2023, NeurIPS 2024, and ICLR 2025 (NeurIPS,

2023; 2024a; ICLR, 2025). Leading laboratories have established dedicated interpretability and alignment teams (OpenAI, 2025; Anthropic, 2025; Google DeepMind, 2025). Governments have created AI Safety Institutes across multiple jurisdictions (UK Government, 2023; NIST, 2023; IAPS, 2025). These developments suggest a new set of concerns is gaining attention. The 1956 Dartmouth proposal (McCarthy et al., 1956) asked: Can machines exhibit intelligent behavior? The emerging question appears to extend this: Can machines exhibit intelligent behavior such that they are aligned, interpretable, and safe? We refer to this as the such-that question.

### 4.2. Fundamentally Different from the Original Question

The founding question of artificial intelligence—whether machines can exhibit intelligent behavior— was deliberately vague, but it admitted an epistemological bypass that made rapid progress possible. Researchers could set aside the philosophically fraught question of what intelligence *is* and instead measure what systems *do*. Russell and Norvig's textbook codifies this move: the field studies rational agents operating in task environments where success criteria can be externally specified (Russell & Norvig, 2021). Whether a bird appears in a photograph admits ground truth. The rules of Go are fixed before play begins. These are what we might call *world-side* problems: the ground truth exists independently of who evaluates the system, the task environment is characterizable in advance, and closed-loop benchmarking becomes a natural methodology. The contrast we draw is not between objective and subjective questions—both kinds admit precise formulations—but between problems whose success criteria are fixed by the task and those that must be constructed through deliberation.

Such-that clauses invert this structure. When we ask whether a system behaves *such that* it remains aligned with human values, interpretable to human understanding, or safe under deployment, we are not asking whether it maps inputs to outputs according to some function waiting to be discovered in the world. We are asking whether its behavior accords with standards that are contested, contextual, and constantly renegotiated through social processes. The ground truth of interpretability or alignment with society does not exist in nature; it must be constructed through human deliberation. Gabriel's analysis of alignment identifies a "simple thesis" that technologists sometimes assume, that one can solve the technical problem first and specify normative criteria afterward, and shows why it fails: normative and technical dimensions are entangled from the start (Gabriel, 2020). The such-that problem is *human-side*, not world-side, and the difference is not one of degree. Here we focus specifically on the normative subset of such-that clauses: alignment, interpretability, safety, governance. Operational constraints like energy efficiency or latency remain largely world-side and benchmark-tractable. The distinction matters because our argument targets the former, not the latter.

Q1 and Q2 are both design problems, but they differ in whether the objective can be specified in advance (Simon, 1996). For Q1, success criteria are externally fixed and agreed upon before the system is built. For Q2, they are not: there is no single accepted definition of alignment, interpretability, or safety. Gabriel documents multiple conflicting conceptions of alignment alone (Gabriel, 2020); the interpretability literature lacks a settled account of what interpretability requires (Lipton, 2018); and fairness criteria have been shown to be mutually incompatible under reasonable assumptions (Chouldechova, 2017). A field that cannot agree on what it is trying to achieve cannot validate that its benchmarks track the goal. Benchmark culture applied to pre-paradigmatic problems does not accelerate progress toward the goal; it accelerates progress toward whichever operationalization happens to gain traction first. We have built technology that outpaces the conceptual vocabulary required to understand it. Hewitt, Geirhos, and Kim argue that we cannot even describe AI behavior using existing human language; interpretability itself is a communication problem requiring the invention of new concepts (Hewitt et al., 2025). The such-that problem demands not just better engineering but something closer to a new science, one for which the methods, and perhaps even the words, do not yet exist.

Recent work from within the machine learning community confirms this diagnosis. Fisher et al. argue that political neutrality in AI is "theoretically impossible" because neutrality is inherently subjective. What seems neutral to one perspective appears biased from another (Fisher et al., 2025). Their framework draws explicitly on philosophy, political science, and sociology, fields that have grappled with neutrality and bias for decades. Rahwan et al. call for a new interdisciplinary field of "machine behavior" that would study AI systems using methods from biology, psychology, economics, and anthropology (Rahwan et al., 2019). These are not peripheral voices; Fisher et al. received an oral presentation at ICML 2025, and Rahwan et al. published in *Nature*. The message from inside the community is increasingly clear: the such-that problem cannot be solved with the tools that solved the capability problem.

### 4.3. The Case for Exaptation

If the such-that problem is genuinely different in kind, one might expect the field's response to differ accordingly. It has not. Current attempts to benchmark alignment and safety (refusal rates, bias metrics, RLHF/RLAIF scores, Constitutional AI evaluations) represent genuine progress. Yet they face a structural difficulty: we are optimizing proxies before

validating that they track the underlying goals. For capability research, decades of experience built confidence that benchmark performance correlated with real-world progress. For such-that problems, this correlation is precisely what remains unestablished. The risk is not that safety cannot be measured. The risk is that premature convergence on current metrics forecloses discovery of what we actually need to measure. The concepts required to measure it may not yet exist.

History offers at least one suggestive precedent. The algorithm now called backpropagation, foundational to modern deep learning, did not originate within artificial intelligence research. Its mathematical core, the reverse mode of automatic differentiation, first appeared in Linnainmaa's 1970 master's thesis on numerical error analysis, written in Finnish with no reference to neural networks (Linnainmaa, 1970). Related ideas appeared in optimal control theory (Kelley, 1960). Werbos, whose 1974 PhD thesis applied these techniques to neural network training, was working not in computer science but in the behavioral sciences, attempting to model political forecasting (Werbos, 1994). Rumelhart, Hinton, and Williams popularized the method for machine learning over a decade later (Rumelhart et al., 1986). The lesson is not that "the solution was waiting." Considerable conceptual work was required to recognize the relevance and bridge the gap. The deeper lesson is that the field could not have specified in advance what it needed, and therefore could not have benchmarked toward it. Someone had to be working at the boundaries, in contact with multiple traditions, for that bridge to be built at all.

What might the analogous story look like for such-that problems? We do not know. Neither does anyone else. Bengio et al. state explicitly that we should search for technical guarantees of safety if they exist, while acknowledging that we do not know whether such guarantees exist (Bengio et al., 2024). This is not ordinary uncertainty about parameters. It is uncertainty about whether the problem admits the kind of solution we are searching for. Under such conditions, the exaptation tax becomes acute. Benchmark-driven selection may be optimizing confidently in a subspace that does not contain what we most need to find, while filtering out exploratory work from which unexpected solutions have historically emerged. The such-that problem may require innovations in methods, but also in measurement regimes, conceptual vocabulary, and institutional structure, domains where benchmark-driven selection has less purchase.

Benchmarking itself was an institutional innovation that enabled the capability era. The Common Task Framework emerged from evaluation programs designed for specific government needs, then was repurposed as a general selection mechanism for ML (Koch & Peterson, 2024). But benchmarks harvested what prior exploration had planted. Neural networks persisted through two AI winters despite a selection environment that had largely moved on, sustained by researchers who stayed at the boundary as connectionism became unfashionable and institutional support shifted elsewhere. The deep learning boom required the selection mechanism and something to select from. For capability research, the accumulated exploration capital was sufficient. For such-that problems, we cannot assume the same. The concern is that if benchmark culture crowds out exploratory work, the next era will have nothing to exapt.

## 5. Proposals

The proposals below outline institutional mechanisms for restoring exaptive capacity without abandoning benchmarking: pluralizing selection criteria, protecting exploratory communities, and governing the selection rules themselves.

### 5.1. Plural Evaluation Regimes

Peer review in AI is nominally multi-criteria but decisions are often made by collapsing those criteria into a single ranking. When commonly used criteria such as novelty, correctness, and empirical validation pull in different directions, aggregation obscures what kind of contribution a paper actually makes. An alternative is to run multiple scoring regimes in parallel (e.g., one that highly rewards novelty and another that rewards benchmark performance) and to admit a portfolio that includes top papers from each regime. Under such a system, a paper that introduces a new approach but does not improve benchmark performance could still be ranked highly in one regime and published. Venues such as TMLR already instantiate part of this logic, evaluating submissions on significance and correctness rather than novelty thresholds or benchmark gains (Transactions on Machine Learning Research, 2022). The such-that problem demands more: venues that explicitly protect work whose goals are themselves under construction.

A complementary mechanism is explicit diversification combined with controlled randomness once a basic quality threshold is met. Peer review in AI is noisy with substantial variance across reviewers (Fogelholm et al., 2012; Aczel et al., 2025). Under time pressure, reviewers often overweight easily checked signals such as benchmark performance, which can distort judgments about criteria more relevant to exaptive capacity (Alberts et al., 2014; Severin & Egger, 2021). When many reviewers apply the same shortcut, the resulting bias becomes correlated noise, and partial randomization among near-equal submissions can improve selection (Kleinberg & Raghavan, 2018; Emelianov et al., 2020). Similar logic underlies modified lottery schemes used in research funding (Adam et al., 2019; Liu et al., 2020; Heyard et al., 2022) and structured hiring interven-

tions such as the Rooney Rule, which enforce diversity at the consideration stage (Kleinberg & Raghavan, 2018; Celis et al., 2021). Stochastic selection is a standard tool for maintaining diversity and avoiding premature convergence in evolutionary computation and search theory (Lehman & Stanley, 2011; Emelianov et al., 2020).

Plural regimes should also treat disagreement as signal and not failure. High variance in assessments often indicates that a contribution challenges existing problem framings or relies on premises not yet widely shared (Stanley & Lehman, 2015). An idea that appears valuable to some experts but not others is not a failed evaluation; it marks a frontier where prevailing frameworks have not yet stabilized. Preserving such disagreement allows ideas with latent potential to persist long enough to realize it.

Notably, a similar pluralism has already emerged within AI systems themselves (Sorensen et al., 2024). Contemporary architectures increasingly rely on ensembles, routing, and specialized components, with overall performance arising from coordination rather than dominance by a single model. The same community has developed novelty search, quality diversity algorithms, and uncertainty-aware exploration methods precisely because pure objective optimization converges too fast and too narrowly (Lehman & Stanley, 2011). Formal models of scientific communities confirm the tradeoff: more connected networks reach consensus faster, but less connected ones are more likely to reach the correct answer (Zollman, 2007). Research evaluation faces an analogous problem but applies the simplest possible method to a task its own algorithms would approach very differently.

## 5.2. Protecting Exploratory Work and Enabling Paradigm Turnover

Plural evaluation should be complemented by explicit mechanisms that protect communities with distinct standards of progress. Ideas that challenge dominant assumptions rarely perform well under prevailing metrics at early stages. Without protected environments in which alternative criteria are legitimate, such ideas are eliminated before they can mature. Institutions should therefore support venues, funding streams, and career signals that allow exploratory work to persist even when misaligned with mainstream benchmarks.

In fast-growing fields, progress is often limited not by a lack of new ideas, but by the slow displacement of dominant frameworks that continue to perform well on existing metrics (Kuhn, 1997; Foster et al., 2015). Review and funding processes should explicitly value work that stress-tests or falsifies core assumptions, even when it does not offer an immediate replacement or benchmark improvement.

Selection mechanisms must also account for the fact that incremental advances within a paradigm compose cleanly while shifts in paradigm typically require coordinated changes across architectures, objectives, training methods, and evaluation itself. Institutions should therefore tolerate incomplete or uneven contributions that make sense only as part of a broader reconfiguration, rather than penalizing them for failing to deliver isolated gains.

Hochreiter and Schmidhuber published the LSTM in 1997 and watched the field move to support vector machines (Hochreiter & Schmidhuber, 1997). The paper sat slow to accumulate citations through a decade of benchmark-driven dominance. It later powered the voice assistants that reached production at scale and sustained the sequence modeling tradition that made the Transformer possible. The work this paper is trying to protect does not announce itself. It looks, from the outside, like the LSTM looked in 2003: real, rigorous, and invisible to the metrics that matter. If you are working on a question the field cannot yet evaluate, you are not in the wrong place. You are in the only place from which certain kinds of progress can come.

## 5.3. Reflexive Governance of Evaluation Criteria

In addition to pluralizing selection and protecting exploratory communities, the research ecosystem must maintain the capacity to interrogate and revise its own selection rules. Evaluation criteria in AI function as the main governance mechanisms: they define what counts as progress and which failures are tolerable. Institutions should therefore legitimize and support meta-level work on evaluation itself, including empirical and conceptual analyses of benchmarks, metrics, and problem framings. Critique of dominant selection rules should be treated as a core research activity with dedicated venues and tracks explicitly designed for such work such as the ICML position paper track. Initiatives of this kind play an important role in making evaluative assumptions explicit and open to revision. The stakes extend beyond the research community: benchmark culture is now being encoded into regulatory frameworks for AI, and evaluation assumptions that harden into legal standards are an order of magnitude harder to revise than those that remain community norms.

Finally, training and norms should equip researchers to recognize selection rules as contingent rather than fixed. This includes mentoring around the practice of reviewing itself, emphasizing judgment about ideas, assumptions, and potential rather than reliance on surface metrics. Conferences can reinforce these norms by recognizing reviewing quality explicitly: ICML 2026 introduced Gold and Silver reviewer designations based on review quality rather than throughput alone. The community should go further: the most rigorous, idea-engaged reviews, those that evaluate a contribution's conceptual ambition and long-term potential, should be citable scholarly contributions in their own right.

# 6. Alternative Views

**The Success Objection.** Benchmark culture has produced AlphaFold, GPT-4, diffusion models, and the modern LLM ecosystem. The field has never been more productive or consequential. Where exactly is the problem? We do not dispute this success, but we question its attribution. The deep learning revolution required benchmarking as a selection mechanism. It also required something to select from. Neural networks persisted through two AI winters despite a selection environment that had moved on: Rosenblatt's perceptron in the 1950s, Hinton and Rumelhart in the 1980s, Bengio and LeCun through the 1990s and 2000s as the community shifted toward SVMs despite available connectionist results. Benchmark-driven selection has harvested more than it has planted. For capability research, the question of whether machines can exhibit intelligent behavior, benchmark-driven selection has been extraordinarily effective at harvesting accumulated exploration capital. Our argument is forward-looking: the selection regime optimized for Q1 may not be optimized for Q2, the question of whether machines can exhibit intelligent behavior *such that* they are aligned, interpretable, and safe. These are different problems with different success criteria (Section 4.2). Past performance on capability questions is not evidence of future performance on such-that questions. The exaptation tax is not visible in current achievements; it is visible—or rather, invisible—in the paths not taken that might matter for problems we are only beginning to face. We are not arguing the system failed. We are arguing it succeeded at one problem and now faces another.

**The Invisibility Objection.** If lost innovations cannot be observed, the exaptation tax is unfalsifiable. Every example cited—backpropagation, attention, RLHF—eventually succeeded. We concede the specific counterfactual is unavailable, but this epistemic situation is familiar from conservation biology: one cannot identify which lost species would have proven essential, yet the case for preserving biodiversity under uncertainty is widely accepted. Structural narrowing is observable: thematic diversity in AI research has stagnated, with increasing specialization toward computationally intensive methods (Klinger et al., 2020). We do not claim benchmark culture is the sole cause—industrial organization, compute costs, and convergence on effective methods all contribute—but benchmark-driven selection plausibly amplifies the narrowing by creating increasing returns for a particular resource profile. Moreover, Lehman et al. argue that ML systems face a "Knightian blindspot" when confronted with qualitatively novel challenges (Lehman et al., 2025); the same logic applies to evaluation systems themselves. Under genuine uncertainty about which directions will prove essential, reduced diversity is a cost in expectation—even when specific losses remain unobservable.

**The Protection Objection.** "Protected venues" could shelter unproductive research from accountability. How do we distinguish sleeping beauties from deservedly ignored work? We cannot, and do not claim to. Protection means plural evaluation, not absent evaluation: work must demonstrate excellence under *some* criteria—rigor, conceptual clarity, novelty—even if it does not improve benchmark performance. Some protected work will fail; that is inherent to exploration. The question is whether a diversified portfolio outperforms a concentrated one when we cannot predict which directions matter. We propose time-limited runway and ex-post tracking, not permanent shelter.

**The Industry Lab Objection.** Elite labs already escape benchmark pressure and produce transformative work. The market has self-corrected; these proposals are unnecessary. If the field's exploratory capacity concentrates in organizations with patient capital, two concerns follow. First, which questions get explored depends on which questions those organizations prioritize—a narrow governance structure for a broadly consequential technology. The such-that questions have public-goods characteristics that may be undersupplied by organizations optimizing for competitive advantage. Second, academic research risks losing relevance for the field's most important problems. Our proposals address whether exploratory capacity should be rationed by resources or distributed by design.

**The Measurement Objection.** Such-that problems need *more* rigorous measurement, not less. RLHF succeeded because it could be evaluated. The paper could justify abandoning rigor where rigor matters most. We agree that developing metrics for alignment, interpretability, and safety is necessary. Our argument is not against measurement but against treating metrics as dominant selection criteria before validating that they track our actual qualitative goals. For capability research, experience gave confidence that benchmark performance correlated with progress. For such-that problems, this correlation is precisely what remains unvalidated. The risk is mistaking measurability for validity—optimizing legible proxies while true objectives remain underspecified. Metrics are tools; benchmark *culture* is a selection regime, not one we endorse extending where the metric-objective gap is not yet understood.

# 7. Discussion

Our argument is not that benchmarking is wrong but that the current calibration is off: the field is over-exploiting a selection regime optimized for Q1 at a moment when Q2 demands more exploration. But the asymmetry of error costs matters. If we are wrong, the cost is some inefficiency. If we are right, the field may be filtering out what it most needs to find. Our proposals are not alternatives to benchmarking. They are hedges against a future we cannot yet evaluate.

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
