# OpenReview forum: "Position: There are futures that benchmark-driven AI cannot see"
_ICML.cc/2026/Position_Paper_Track — ICML 2026 Position Paper Track spotlight_

### Official Review · Reviewer_8S1b · 2026-02-19

**Significance:** 4
**Argument Clarity:** 3
**Rating:** 6
**Confidence:** 5

**Questions:**

Please see the weakness section.

**Alternative Views Section:**

Yes

**Compliance With Llm Reviewing Policy A Conservative:**

Affirmed.

**Discussion Potential:**

4

**Final Justification:**

This paper is pretty solid. I keep my recommendation for Strong Accept.

**Paper Summary:**

This paper argues that the benchmark-centered environment in AI reorganizes research environment and agendas. This shift is not incidental to a narrowing in research questions, but it **produces** the narrowing. The argument is established through the lens of **exaptation**, which is a term in biology denoting that a feature evolved for one purpose but later became instrumental to another (unforseeable) function. Exaptation not only happens in biological evolution, but also applies to evolution, maturation, and adaptation of ideas in the scientific community. Exaptation in science crucially depends on long-term survival of ideas whose value is not yet legible. The benchmark-centered culture is **detrimental to such long-term survivorship**, because the selective pressures structurally organize (i) who gets to participate, (ii) what existing participants choose to work on, and (iii) what gets recognized vs. self-disqualified.

This paper proposes three future actions:

**1. Diversify selection criteria**, via either parallel scoring regimes, or controlled randomness

**2. Protect exploratory research**, via tolerating incompleteness of works that stress-test or question core assumptions, without pre-maturally eliminating them.

**3. Govern selective rules**, via dedicating track to meta-research on evaluation and critiques of dominant selection rules, as well as recognizing reviews as scholarly contributions.

**Position:**

Yes

**Position In Title:**

Yes

**Related Work:**

4

**Strengths And Weaknesses:**

### Strengths

1. This paper synthesizes **a wide array of literature**, spanning science of science, evolution, critics of ML scholarship, as well as a branch of sociology/economy that studies scientific communities. Together, they well support the premise, exposed worries, and implications drawn from them.

2. The argument does not end at how benchmarks hinders exaptation, but **deepens to illuminate an acuter challenge grounded in what the authors call the ***such-that*** problem**. The such-that problems marks a shift of focus from capability (whose ground-truth is easier to find in nature and be measured against), to normativity (whose ground-truth is a result of social negotiation and must be persistently contested). The such-that problem renders benchmarks even less adequate on its own, thus providing a stronger motivation to incorporate diverse sub-communities.

3. The **“it-harvests-more-than-it-plants”** argument is **particularly compelling**! It is not only provides a comprehensive view of historical inquiries, but it is also a nice rebuttal to “benchmarks are successful so we should keep doing it”. The “it-harvests-more-than-it-plants” argument reveals a deeper source of benchmarks’ success which is not due to themselves, but due to an earlier exploratory capital. Only in this sense are benchmarks successful. However, benchmarks are **unsuccessful in planting** exploratory capital for the next round.

### Weaknesses:

I really enjoyed reading this paper and I do not see major weaknesses. Perhaps two minor suggestions:

**1. Can you make the subsections in $\S3$ and $\S4$ better align?**

Let me elaborate. I can see a two-stage structure of this paper. Even if you don’t bring up the such-that problem, you will have enough material to write a paper about how the benchmark culture imposes an exaptation tax. And you can justify this by illustrating how benchmarks organize (i) who gets to participate, (ii) what current participants gets to do, (iii) what gets recognized vs. self-disqualified, all at the cost of exaptation. $\S3.1, 3.2, 3.3$ corresponds to these (i), (ii), (iii). This is stage1.

The such-that problem is stage2, which makes every problem you just raised **acuter**. I think it would be nice to make $\S4.1, 4.2, 4.3$ align to $\S3.1, 3.2, 3.3$, that is,

$\S4.1$ how does the such-that problem make (i) acuter?

$\S4.2$ how does the such-that problem make (ii) acuter?

$\S4.3$ how does the such-that problem make (iii) acuter?

I understand that you might want to highlight that it’s not merely a heightened *degree*, but a new *kind* of problem. And “acuter” might obfuscate this distinction. So my phrasing might need refinement. I trust you to think of a way to achieve both structural alignment in the manuscript and emphasis on the capability $\rightarrow$ normativity shift as a different kind rather than degree.

**2. Can you add a call-to-action for individual researchers/research-groups?**

The three proposals in $\S5$ target at large parties — institutions, conferences, funding agencies. They are great! I didn’t mean to critique them.

I think it would be nice to propose something that target at small parties. For example, regarding the measurement problem in safety, the motivation is clear that we want the operationalizable proxies to track what we really want to measure. But it is unclear *how* to do that. We lack methodology, or even a conceptual framework to think about such problems. It would be helpful to cite emerging works that confront this problem and attempt at building conceptual or methodological frameworks. Broadly, it would be greatly helpful if you can **cite emerging works that precisely exemplify the type of work $\S5.2$ aims to *protect*.** This will give researchers who struggle in the current benchmark culture more confidence and some solid lines to follow.

**Support:**

4

---

> ### Author Rebuttal · Authors · 2026-03-31
>
> Thank you for the careful reading and the structural suggestion, which was useful to think through.
>
> On the alignment between Sections 3 and 4: we considered restructuring Section 4 to mirror the three dimensions of Section 3 explicitly, but this risks forcing artificial parallelism onto what is a different kind of argument. The such-that problem is not a heightened version of each dimension but something new in kind. We will add a bridging paragraph at the opening of Section 4 in the camera-ready that makes the connection back to the three dimensions explicit without collapsing that distinction.
>
> On the call-to-action for individual researchers: we will add a closing note in Section 5 oriented toward engaging with the conceptual and measurement gaps the such-that problem exposes. We have deliberately avoided anchoring this to specific emerging works, since doing so would itself enact a version of the premature convergence the paper warns against.

---

> > ### Author Rebuttal · Reviewer_8S1b · 2026-04-01
> >
> > The rebuttal adequately addressed my questions.

---

### Official Review · Reviewer_vYco · 2026-02-26

**Significance:** 3
**Argument Clarity:** 3
**Rating:** 6
**Confidence:** 4

**Questions:**

I do not want to nitpick on every point raised by the paper as it is overall well articulated but there are some points that perhaps could be more nuanced.

For instance, the paper says that
> Neural networks persisted through two AI winters, sustained by researchers working without benchmark validation: Rosenblatt in the 1950s, Hinton and Rumelhart in the 1980s, Bengio and LeCun through the 1990s and
2000s when connectionism was unfashionable.

which does not feel right to me LeCun work was also driven on MNIST accuracy, the early work of Bengio also reported performance on some metrics for text modelling, etc.

> Critique of dominant selection rules should be treated as a core research activity with dedicated venues and tracks explicitly designed for such work such as the ICML position paper track.

I feel that this is already the case so perhaps the wording could be changed as "should be more valued", for instance, the paper "Deep reinforcement learning at the edge of the statistical precipice" which receives the Outstanding Paper Award from Neurips comes to mind but there will also be lots of others that question some metrics or some protocols.

>  The attention mechanism emerged from computational linguistics

It would be good to support this claim with a reference. I often see Bahdanau or Graves being credited to introducing the modern form of attention.

I observed some typo with spacing on citation:
* l39 Russel
* l95 Russel
and a few other places in the paper


While reading your paper, I was thinking about the review of the famous "Generative Adversarial Nets" paper https://papers.nips.cc/paper_files/paper/2014/file/f033ed80deb0234979a61f95710dbe25-Reviews.html

where the authors we critized for their empirical results and writing the following despite the work that ended up being groundbreaking:
> "While we make no claim that these samples are better than samples generated by existing methods, we believe that these samples are at least competitive with the better generative models in the literature and highlight the potential of the adversarial framework."

The paper got accepted but the decisions appears borderline which may be related to the point you are making about over-indexing incremental gains on benchmarks despite the quality of ideas.

**Alternative Views Section:**

Yes

**Compliance With Llm Reviewing Policy A Conservative:**

Affirmed.

**Discussion Potential:**

4

**Final Justification:**

The rebuttal adressed all my concerns, I have raised my score accordingly.

**Paper Summary:**

The paper discuss the impact of the benchmark driven culture on AI. It describes AI history in details together with the concept of exaptation: some ground-breaking methods originated from "dormant" ideas which may not have obtained very good performance on benchmarks but ends up bringing massive improvements once conditions are met. It then discusses the limits of such approaches which appears essentially when we ask questions which may be hard to assess (presently) with benchmarks such as AI safety or interpretability (question which authors call "such-that problems"). Those are different since they require for instance alignment of values which is currently hard to translate to a single benchmark (because values are subjectives etc). It also describes the exaptation tax, i.e. what is being lost by restricting exploration due to a selection process driven by empirical improvement which selects/favors incremental methods. Finally, alternate views are addressed discussing potential arguments to the paper thesis.

**Position:**

Yes

**Position In Title:**

Yes

**Related Work:**

4

**Strengths And Weaknesses:**

Strengths:

The paper is extremely well written. It is very pleasing to read and goes into depth to explain its position. The argumentation is clear and while the position is assumed, the paper does a great job when discussing alternate views with the same rigor than when presenting its thesis.

The paper is well referenced. The history of AI is well transcribed and in particular in clear relation to the thesis of the paper.

The remediation proposed by the paper are sound and also well argumented.

Weaknesses:
The articulation with the Such-that problem section 4. felt a bit weaker to me. They are ways to evaluate safety and alignement for instance although perhaps this is not as clearly defined as other benchmarks mentioned it feels to me that those are also benchmarked and improved to a good extent? (I get your point that some aspects requiring value alignment may be less suited but it does not seem to apply to all categories listed here?) This is also tied with exaptation discussion which makes sense but feels a bit orthogonal to me?

While counterpoints are well written, I felt that some efforts done by the community to improve on not over selecting with benchmark incremental performance could be noted. For instance, TMLR is one venue explicitly allowing research to be presented even if not seen as not novel enough but interesting for the community [1]. A related point too, the writing felt to me as presenting the choice as binary (you either follow the benchmark culture or you dont) whereas it feels to me much more like a continuous trade-offs balancing exploration and exploitation (as in genetic algorithms mentioned in the paper). In that sense, I feel the discussion is more on how calibrated this trade-offs is (I agree with the paper position that it may be over exploiting).

[1] https://jmlr.org/tmlr/acceptance-criteria.html

**Support:**

4

---

> ### Author Rebuttal · Authors · 2026-03-31
>
> Thank you for several points that strengthen the paper.
>
> The reviewer is right about LeCun and Bengio. Both worked within benchmark regimes and we will correct the sentence in the camera-ready. The broader point still stands: the community moved toward SVMs despite available results, and LeCun, Bengio, and Hinton noted this themselves in their 2015 Nature paper.
>
> On the binary framing, we agree. The Recht quote in Section 3.2 already frames this as calibration rather than opposition. We will make it explicit in the Discussion in the camera-ready: the argument is about recalibrating the exploration-exploitation balance, not abandoning benchmarking.
>
> On whether such-that problems are already benchmarked: they are, and we do not claim otherwise. Refusal rates, bias metrics, RLHF scores represent real progress. Our argument, which we will sharpen in Sections 4.2 and 6, is that the metric-objective gap for such-that problems is fundamentally different from capability research. For capability benchmarks, decades of use built confidence that performance tracks real progress. For such-that problems, we are optimizing proxies before that confidence has been established. Our concern is that benchmark-driven selection may foreclose the exploratory work these goals most require.
>
> We will also add a brief mention of TMLR in Section 5.1 as an existing instantiation of plural evaluation worth building on, update Section 5.3 to acknowledge existing progress while arguing for more, add Bahdanau et al. (2014) at the opening sentence of the attention paragraph in Section 3.1, and correct the Russell typo throughout.

---

> > ### Author Rebuttal · Reviewer_vYco · 2026-04-02
> >
> > The minor points I raised were adressed. Regarding my other points, I believe that a few sentence as proposed by the author will allow to clarify my other points. I have raised my score to reflect this.

---

### Official Review · Reviewer_qv7c · 2026-03-06

**Significance:** 4
**Argument Clarity:** 4
**Rating:** 6
**Confidence:** 4

**Questions:**

none

**Alternative Views Section:**

Yes

**Compliance With Llm Reviewing Policy A Conservative:**

Affirmed.

**Discussion Potential:**

4

**Final Justification:**

I'm very happy with the paper. It is thought-provoking and timely. Since there was not much discussion needed, there is not much to say here. It is by far the most interesting position paper in my assigned batch. I'm only hoping that the authors will revise the sections I commented on as they promised.

**Paper Summary:**

The authors argue that there is too much emphasis on benchmarking for selecting what should get attention and what should not. To this end, they bring up several historical examples and theories to support their position that selection based on benchmarking "imposes an exaptation tax". This means that we do not know which technique or approach might be important in the future to further improve AI -- so we cannot easily discard proposals. The authors propose several ideas to mitigate the problem, incl. more multi-objective selection of papers, protection of alternative exploration paths, and regular revision of their own selection criteria.

**Position:**

Yes

**Position In Title:**

Yes

**Related Work:**

4

**Strengths And Weaknesses:**

Strength:

* The position of the authors was discussed in other contexts, but it is very timely to rethink this with the recent breakthroughs and benchmarks of AI in mind. It is a thought-provoking position that allows a lot of discussion.
* The paper is very well written and is easy to read (incl. nice examples), and at the same time delves deep into several aspects of AI benchmarking
* There are clear proposals of how the community could deal with the "problem"
* The alternative views are convincingly stated and show how they are *not* contradicting the position of the authors.
* The literature seems to be very well covered (I'm not an expert), both fairly old papers and new papers related to the topic or examples of related topics.

Weaknesses:
* At times, I had the impression that the authors indirectly argue against aligned, interpretable and safe AI. I don't think that the authors really have a problem with that -- or at least I hope so -- but I believe that this should be clarified. For example, I'm not sure whether the authors have a problem with the ethics reviews of NeurIPS (Section 4.1).
* Some typos with whitespaces or punctuation

**Support:**

4

---

> ### Author Rebuttal · Authors · 2026-03-31
>
> Thank you for the careful reading.
>  On the concern that Section 4.1 might read as arguing against aligned, interpretable, and safe AI: this is the opposite of our intent. The structure of Section 4 makes the arc visible from its subsection titles: 4.1 observes that the such-that problem is already recognized at the center of the field; 4.2 argues it is fundamentally different in kind from the capability question; 4.3 draws the implication for exaptation. The citations to NeurIPS ethics requirements, safety workshops, and alignment teams at major labs are evidence that this shift is real and underway, not targets of criticism. We will add a clarifying sentence in 4.1 in the camera-ready to make this read unambiguous for any reader entering the section without the full arc in mind.

---

> > ### Author Rebuttal · Reviewer_qv7c · 2026-04-01
> >
> > Thanks for the reply. With the promised revision, I'm happy if the paper gets accepted.

---

### Decision · Program_Chairs · 2026-04-30

**Decision:**

Accept (spotlight)

**Comment:**

Based on the reviews and rebuttal, this is a clear accept.

There is general consensus among the reviewers that the position in the paper is very well articulated and it raises timely questions that need to be discussed within the community.

While the reviewers asked to nuance or better clarify some of the positions in the paper (e.g., Rev. qv7c about the position of the authors wrt to safety/aligned AI or the need to better acknowledging some efforts from the community as pointed out by Rev. vYco), the authors did provide convincing arguments in the rebuttal that I would strong encourage to integrate in the final revision.